# Learned Semantic Index Structure Using Knowledge Graph Embedding and Density-Based Spatial Clustering Techniques

**Yuxiang Sun** [1], **Seok-Ju Chun** [2] **and Yongju Lee** [1,*]

1 School of Computer Science and Engineering, Kyungpook National University, Daegu 41566, Korea; syx921120@gmail.com
2 Department of Computer Education, Seoul National University of Education, Seoul 06639, Korea; chunsj@snue.ac.kr
* Correspondence: yongju@knu.ac.kr; Tel.: +82-10-3532-5295

**Abstract:** Recently, a pragmatic approach toward achieving semantic search has made significant progress with knowledge graph embedding (KGE). Although many standards, methods, and technologies are applicable to the linked open data (LOD) cloud, there are still several ongoing problems in this area. As LOD are modeled as resource description framework (RDF) graphs, we cannot directly adopt existing solutions from database management or information retrieval systems. This study addresses the issue of efficient LOD annotation organization, retrieval, and evaluation. We propose a hybrid strategy between the index and distributed approaches based on KGE to increase join query performance. Using a learned semantic index structure for semantic search, we can efficiently discover interlinked data distributed across multiple resources. Because this approach rapidly prunes numerous false hits, the performance of join query processing is remarkably improved. The performance of the proposed index structure is compared with some existing methods on real RDF datasets. As a result, the proposed indexing method outperforms existing methods due to its ability to prune a lot of unnecessary data scanned during semantic searching.

**Keywords:** semantic search; learned semantic index; knowledge graph embedding; linked open data; clustering techniques

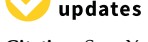



## 1. Introduction

Knowledge graph embedding (KGE) has emerged as a powerful deep learning technology that has achieved remarkable success in artificial intelligence. However, its techniques are rarely employed for semantic web applications [1]. Linked open data (LOD) have been rapidly growing over the past years, consisting of 1301 machine-readable datasets with 16,283 links as of May 2021. The resource description framework (RDF) is a graph-based data model for LOD. The RDF Schema (RDF-S) provides mechanisms for describing groups of related resources and relationships between these resources. Web ontology language (OWL) is a semantic annotation markup language developed as a vocabulary extension of RDF. The same instances in different datasets are interlinked with owl:sameAs, which is a built-in OWL property. To the best of our knowledge, the research on how semantics are embedded and used to train neural network models to achieve instance matching in LOD is relatively meagre.

Although numerous state-of-the-art methods are applicable to the LOD cloud, there are still many ongoing problems in this area. One challenge is the interaction between applications using different vocabularies or ontologies. Some annotations face the heterogeneity problem because no standard labelling method applies to all datasets. This problem can occur when identical entities are represented in different ways in multiple datasets. For example, "Benz E-250" and "Mercedes Benz E class 250" have the same semantics but cannot be matched because they do not belong to the same synonym set. Consequently, an efficient method is needed for semantic searching. Owing to the massive size of the LOD

cloud, it is practically impossible to perform this operation manually. Therefore, a semantic method for automatic classification and discovery is necessary.

Word embedding is a strategy to efficiently represent natural language words in vector space. The purpose of word embedding is to obtain vector values corresponding to each word. Hence, similar words are located at closer distances. This method can be effectively applied to question answering or recommendation systems. Many studies are being conducted for effective word embedding, such as NNLM [2], Word2Vec [3], and GloVe [4]. Word embedding adopts a preprocessed corpus of text, whereas LOD are stored in the form of RDF graphs. Thus, the LOD cloud must use the graph embedding technique instead of word embedding. The RDF2Vec [5] is a method for graph embedding that extracts RDF sequence data and uses the Word2Vec model to embed entities in a low-dimensional space. However, embedded vectors from RDF2Vec are not directly applied to the semantic search.

Although the semantic web community has achieved tremendous success by the LOD project, many researchers argue that semantic technologies have little impact on the real world [6]. In this regard, the KGE technology provided by Google is highly regarded as an example combining concepts and practicalities of the semantic web. The knowledge graph represents the relationships between the head and tail nodes in the form of <*h*, *r*, *t*>, where *h*, *r*, and *t* are the head, relation, and tail, respectively. Here, we must consider how the nodes and relations can be represented as vectors. The TransE model [7] proposed that the sum of the head and relation vectors should be maximally equal to the tail. Therefore, the purpose of the score function is to make $h + r = t$. If all triples satisfy this condition, the nodes and relations can be embedded.

The TransE model shows excellent performance in 1:1 relations, but problems are raised in cases such as 1:N, N:1, and N:N. The TransH model [8] solved the TransE problem by projecting the relations onto the hyperplane. In practice, however, entities (head, tail) and relations are different objects, so the TransR model [9] proposed an idea to embed entities and relations in separate spaces. The ConvE model [10] performed a 2D convolution on embedding vectors and applied nonlinearity to enhance the vector representation. Recently, significant progress has been achieved in applying KGE to entity matching and link discovery tasks [11]. In this article, we propose a novel learned index structure using KGE and spatial clustering techniques for semantic search. The contributions of this study are as follows:

- We propose a learned index structure of the LOD cloud using KGE models, and algorithms of join query processing based on our index structure are discussed.
- The LOD stored in the form of RDF graphs can be automatically classified using the embedded model and density-based spatial clustering algorithm.
- Embedded vectors can quickly be matched to semantically similar clusters by comparing vector similarity between a given query and cluster centroids, thereby significantly reducing irrelevant traversal for complex semantic search.
- Experiment results verify that this improved representation outperforms the state-of-the-art LOD index structures.

The rest of this paper is organized as follows. In Section 2, we describe related work. In Section 3, we present a new learned semantic index structure for semantic search. In Section 4, the experimental evaluation is described. Finally, in Section 5, conclusions are drawn, and future research direction is suggested.

## 2. Related Works

### 2.1. Indexing Techniques for RDF Data

Traditional RDF indexing techniques only focus on optimizing index structures to improve retrieval efficiency. There are two main strategies for RDF indexing—reducing data dimensions and adopting multidimensional index structures. The former mainly uses ordering or transformation techniques to convert RDF data into a lower-dimensional space. The advantage is that the converted data are simpler and provide better storage performance. However, it is necessary to preprocess the raw data before retrieval. In

addition, the entire index structure needs to be refreshed to perform insert or update operations. Therefore, the maintenance cost is quite expensive. By contrast, building a multidimensional index structure to retrieve RDF data is more common and widely used. Multidimensional index structures are mainly divided into the data- and space-partitioning-based index structures, which are subdivided into the feature- and distance-based index structures. Table 1 shows their detailed classification and description.

**Table 1.** Index techniques for RDF data.

| | Dimension Reduction | | Multidimensional Index Structure | | | |
|---|---|---|---|---|---|---|
| | | | Data Partitioning | | Space Partitioning | |
| | Ordering | Transforms | Feature | Distance | Feature | Distance |
| Characteristic | Convert RDF data into lower dimensional space | | Build the multidimensional index structure | | | |
| Advantage | Excellent storage performance | | Efficient query processing | | | |
| Weakness | Need refresh entire index structure when new data are inserted | | High spatial complexity with the increase of data volume | | | |
| Related work | Tentris [12], DISE [13] | DiStRDF [14], TripleID-Q [15] | SPT+VP [16], | axonDB [17], | HTStore [18], | Vp-tree [19] |

### 2.2. Storage Structure for LOD

There are three alternative approaches. The first is maintaining independent data copies in a centralized registry, benefiting from convenient conditions for efficient query processing—called the centralized approach. The second approach is based on accessing distributed data on the fly using a link traversal—called the distributed approach. The distributed approach performs queries over multiple SPARQL endpoints that publishers provide for their LOD datasets. The third is the index approach for efficient query processing over LOD resources. The LOD index structures are similar to traditional database query processing techniques. Existing data summaries and approximation techniques may be adapted to develop an index structure for LOD queries.

We have proposed extended multidimensional histograms (MDH*) for LOD storage and indexing [20]. The MDH* aims to support efficient join query processing with a compact storage layout. The first step of building MDH* is to transform the RDF triples into numerical numbers. These numbers are points in the $n$-dimensional data space, whose coordinates correspond to three-dimensional cubes. The coordinates are inserted sequentially and aggregated into regions. Each region maintains a list of resources. Each resource in the list is added with two additional occurrences to improve the join query performance.

### 2.3. Word Embedding

Embedding means a series of processes that transforms natural languages into machine-understandable vectors, converting words into vectors, and inserting them into vector space. The semantic similarity can be calculated from the spatial distance between vectors. For example, "car/truck/motorcycle" related to vehicle, "pianos/saxophones" related to music, and "chicken/black-bean-sauce/ramen" related to cooking can be clustered and represented as nearby vectors. This is because the purpose of word embedding is to find vector values corresponding to each word so that similar words are located at a closer distance. Various techniques have been developed, such as NNLM, Word2Vec, GloVe, and FastText. In particular, the Word2Vec method was published by Google's research team; its basic structure consists of CBOW and Skip-Gram. The CBOW method uses peripheral words to predict a single target word, whereas Skip-Gram is trained to predict surrounding words. Skip-Gram is more effective than CBOW because it can obtain more learning data from a corpus. Using these techniques, it can be inferred that if words frequently appear

together, they have similar meanings. It is possible to learn to position vectors of words with similar meanings close.

### 2.4. KGE

Existing studies for KGE include TransE, TransH, TransR, and DistMult models [21]. Bordes et al. [7] proposed the TransE model. They treated the predicate in each triple as a mapping from subjects to objects and adjusted the corresponding vectors of head (subject), relation (predicate), and tail (object) to make $h + r = t$. This model has few parameters, low computational complexity, good performance, and scalability on large-scale sparse data. However, the model only applies to a 1:1 relationship and cannot represent 1:N or N:1 relationship.

To address this issue, the TransH model was proposed. The TransH model proposes that an entity has different representations in different relationships. The triangular closure relationship is broken by using a projection. The algorithm does not strictly require $h + r = t$ but only needs to satisfy the projection collinear of head and tail nodes on the relational plane. In addition, tail nodes are replaced for N:1 relationship, and head nodes are replaced for 1:N relationship, satisfying $h_\perp + r = t_\perp$.

The abovementioned two models assume that entities and relationships are embedded in the same space. However, an entity may have multiple properties, hence there are cases where heads, tails, and relations are not in the same space. To solve this problem, the TransR model was designed. Its basic principle is to divide the whole space into entity and relation spaces. The head and tail nodes in the entity space are mapped to relation space via a relation matrix projection operation by projecting entities from entity space to the corresponding relation space and then building translations between projected entities. Figure 1 shows the entity and relation space of the TransE, TransH, and TransR models.

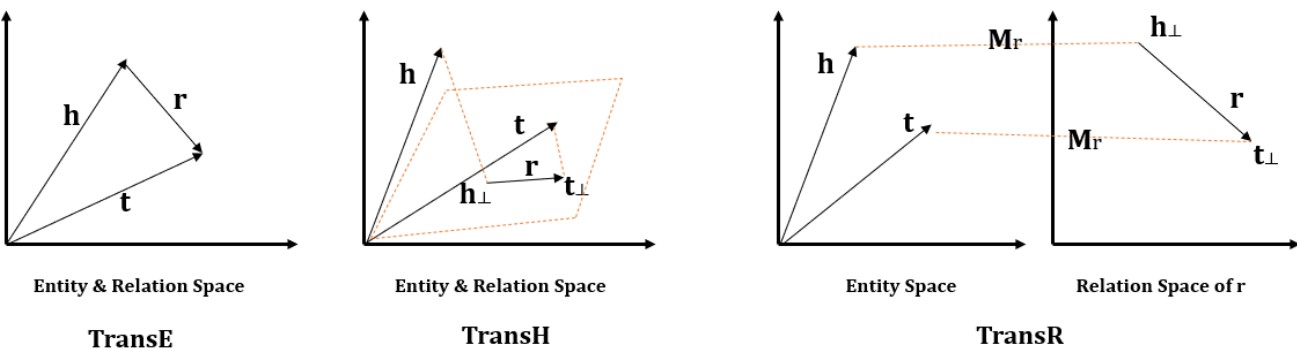

**Figure 1.** Entity and relation space of TransE, TransH, and TransR.

The abovementioned three models can be collectively referred to as translation-based models. Moreover, bilinear-based models, such as RESCAL [22], DistMult, and complex [23], as well as neural network-based models, such as ConvE, have been investigated for deep learning.

## 3. Learned Semantic Index Structure

In this section, we propose a learned semantic index structure using KGE and density-based spatial clustering techniques. Algorithms of SPARQL join query processing based on our index structure are discussed.

### 3.1. Architecture of Learned Semantic Index Structure

We propose a hybrid LOD storage strategy combining the index and distributed approaches. The index approach offers better performance, but the queried data might not be current because LOD has changed considerably. The distributed approach provides recent data, but the query execution is slower because data must be transmitted via the

network. Our proposed approach may solve these problems. By using a learned semantic index structure, our method can retrieve distributed LOD datasets efficiently.

State-of-the-art RDF storage techniques may be adapted to develop an index structure for LOD queries. For instance, Sun et al. [24] proposed a hybrid index structure, which was a combination of R*-tree and *k*-d trees. The goal of this method is to support efficient join query processing without significant storage demand. The core task in building the index is to transform RDF triples into points in a three-dimensional space by applying hash functions (e.g., the hashCode() function in Java) to the individual components of the RDF triples. However, this method considers nothing about the data distribution and does not exploit common patterns prevalent in real-world data. Typically, existing LOD index structures only consider the found relevant resources. Thus, there is no guarantee that the resources actually provide the RDF triples that were originally searched for (i.e., false positives). This is because the hash function does not describe exact coordinates in the multidimensional space. The input of LOD index structures is generally a triple pattern, and they output a list of data resources that potentially contribute to the result. Thus, a large number of resources can contribute to each triple pattern. Because accessing too many resources over the LOD cloud is expensive, we need to investigate a novel index structure that can efficiently process queries over the distributed LOD.

It is not the intention of this paper to discuss which index structure is appropriate for RDF storage. We describe instead an extension of the existing index structures which integrates two index structures for clusters and the centroid of each cluster. In the existing index structures, the most popular R*-tree and *k*-d tree are selected for our index structure. As the cost of developing a new index structure can be more expensive than the cost of simply extending an existing one, adding new features to the existing index structure is an excellent alternative.

Figure 2 depicts a workflow for constructing our learned semantic index structure. Our main idea is to turn heterogeneous hash codes into homogeneous sets. The heterogeneity of indexing keys is the main reason for long query processing time. If keys become homogeneous, query processing time can significantly reduce. Turning heterogeneous indexing keys into homogeneous sets entails semantically embedding numeric vectors in the adjacent embedding space, which we demonstrate to be achievable using KGE techniques.

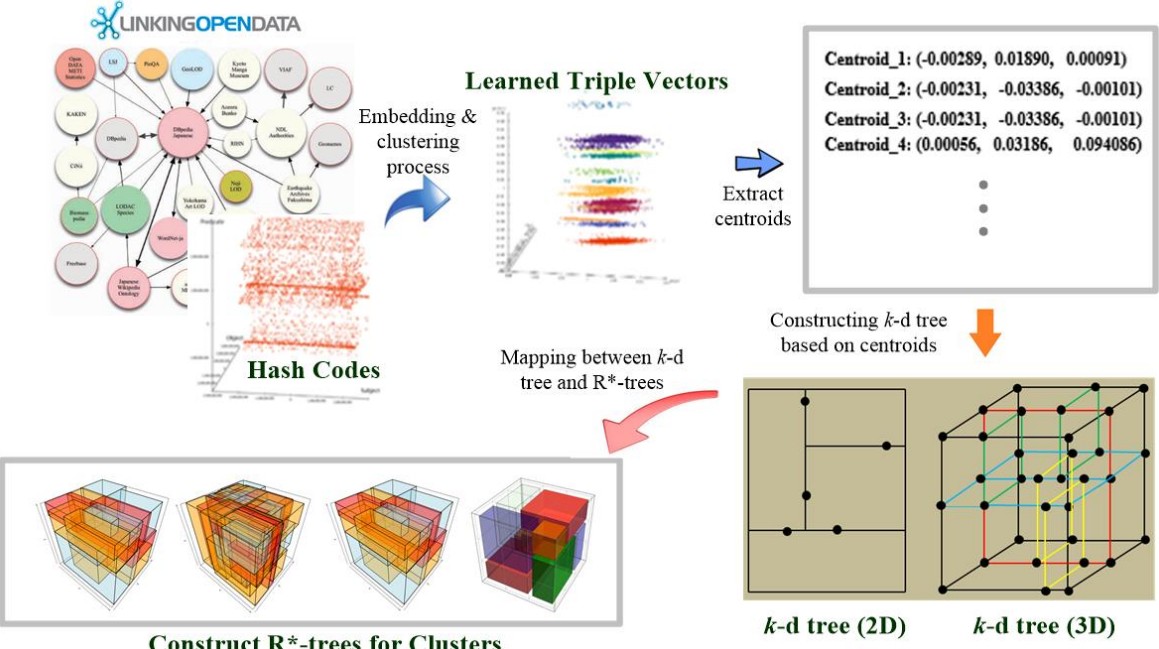

**Figure 2.** Workflow for constructing the learned semantic index structure.

The proposed framework comprises three steps. The first step is to perform embedded processing for LOD datasets to obtain vectors of triples. The projected head and tail nodes are mapped by a project matrix $M(r)$ [9]. With the mapping matrix, the method builds the projected entities:

$$h(r) = hM(r), \ \ t(r) = tM(r) \tag{1}$$

Its score function is as follows:

$$f_r(h, \ t) = ||h(r) + r - t(r)||_2^2 \tag{2}$$

The second step is to cluster vectorized triples and obtain each cluster centroid from a density-based spatial clustering algorithm. Given a set of points in some space, it groups all neighbors within a given radius. The centroid of a cluster is defined as a point in the $n$-dimensional data space found by averaging the measurement values along each dimension:

$$\overline{x_i}(c) = \frac{1}{N_c} \sum_{j \in S_c}^{n} x_{ij} \tag{3}$$

where $S_c$ represents the set of indices of cluster $c$ which contains $N_c$ objects, and a cluster centroid $\overline{x_i}(c) = (\overline{x_1}(c), \overline{x_2}x(c), \cdots, \overline{x_n}(c))$. Finally, R*-trees and a $k$-d tree are built for the clusters and their centroids, respectively. The main task of $k$-d tree is to determine the target cluster to reduce data scanning, whereas that of R*-tree is to quickly search the final results within the target cluster. As all data items in LOD are represented in triples of the form <subject, predicate, object>, the LOD cloud uses the RDF2Vec graph-embedding technique. Before RDF2Vec graph-embedding processing, however, the LOD datasets need to be preprocessed because these datasets are represented in the triple format. To perform RDF2Vec graph-embedding operations, Apache Jena [25] under the OpenKE [26] platform is employed to parse and sequence raw data. Figure 3 describes the overall architecture of our index structure schematically.

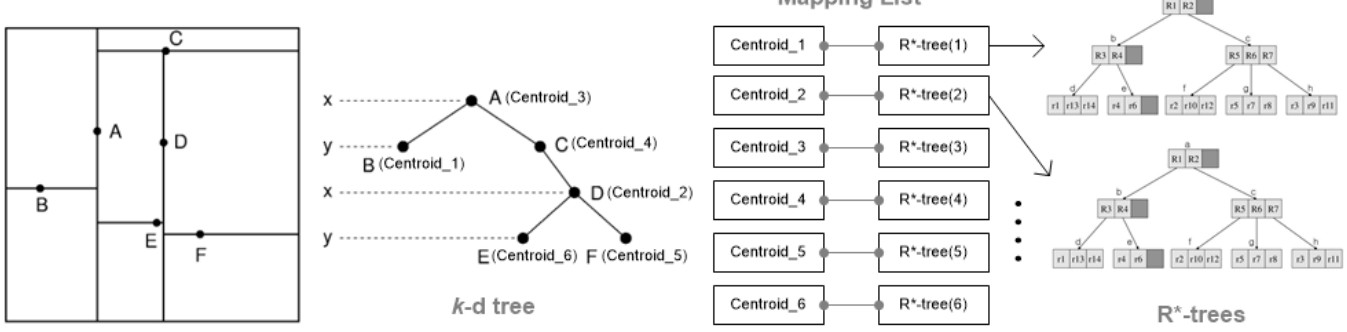

**Figure 3.** Overall architecture of the learned semantic index structure.

### 3.2. SPARQL Join Query Algorithms for Learned Semantic Index

SPARQL is a standard query language for the RDF data model. As SPARQL statements can be represented as graphs, we transform SPARQL into a query graph to perform query processing. There are two types of SPARQL queries—single and join triple pattern queries. Processing queries with a single triple pattern is straightforward. When a query consists of multiple triple patterns that share at least one variable, we call it a join triple pattern query. In SPARQL queries, there are eight triple patterns: (?s, ?p, ?o), (s, p, o), (s, p, ?o), (?s, p, o), (s, ?p, o), (s, ?p, ?o), (?s, ?p, o), and (?s, p, ?o). SPARQL queries can be divided into seven different join types [27]: star (only subject–subject join), chain (only subject–object join), directed cycle (only subject–object), tree (subject–subject and subject–object), cycle (subject–subject and subject–object), single (only subject or object), and complex (random combination).

The nearest neighbor (NN) algorithm finds the nearest vectors to a given query pattern; this is also called similarity or proximity search. Applying this algorithm to high-

dimensional spaces deteriorates performance because the distance between NNs can be large. Thus, much research has focused on finding approximations of the exact NNs. In this study, an approximate NN algorithm is presented to find $k$ approximate nearest vectors in embedding space to a query pattern by using $k$-d tree. We can measure the similarity between each triple vector and a query vector through vector similarity (e.g., cosine similarity and Euclidean distance). Several experiments demonstrated that the Euclidean distance similarity provides excellent accuracy. Therefore, we adopt the Euclidean distance between two vectors in embedding space as the measure of relevance. Let a vectorized triple $v$ and query $q$ have coordinates $(v_1, \dots, v_n)$ and $(q_1, \dots, q_n)$, respectively, in $n$-dimensional embedded space, then the distance $d(v, q)$ between $v$ and $q$ is given by:

$$d(v,q) = \sqrt{\sum_{i=1}^{n}(v_i - q_i)^2} \tag{4}$$

In this study, the density-based spatial clustering of applications with noise (DBSCAN) model was adopted to cluster similar semantic vectors. The DBSCAN model is a well-known, unsupervised machine learning tool for clustering [28]. The DBSCAN creates clusters based on dense regions while marking points that lie alone in low-density regions as outliers. In DBSCAN, the density is detected via core points that are quite sensitive to input parameters: $\epsilon$ is the radius of the neighborhood, and *minPts* is the minimum number of points constrained within $\epsilon$ radius. To ensure that all vectors cluster without noise vectors, minimum samples are set to 1.

There are two key points for complex join queries: join ordering and selecting the most suitable join algorithm [29]. Algorithm 1 shows a detailed illustration of our SPARQL join query algorithm for semantic search. In the process of a query pattern, different join algorithms—hash join, nested loop join, and sort-merge join—are combined to replace the traditional single join query algorithm, because, in the actual experiments, we found that the use of different join algorithms has a certain impact on the actual join query performance. For example, when we want to perform a join calculation on results of two subqueries A and B, using the nested loop join algorithm is suitable if the data volume of A and B is relatively small. However, using the hash join algorithm has better performance if the query result of A is very large. The most suitable query type for hash join is the star query. However, a serious weakness of the simple hash join is the poor performance for complex non-star queries. We propose a hybrid join search algorithm that combines the hash join for star and the nested loop join for non-star queries. In line 2, we judge whether a query (e.g., a complex query type Q = (?s, $p_1$, $o_1$) ⋈ (?s, $p_2$, $o_2$) ⋈ (?s, $p_3$, ?o) ⋈ ($s_4$, $p_4$, ?o)) is one of the possible SPARQL triple patterns. In line 4, the nearest points are calculated from the NN algorithm between each subquery pattern and centroids in $k$-d tree, if the Euclidean distance between them is shortest, the corresponding R*-tree will be searched by the mapping list. In line 6, each subquery's search results will be added to the result list. From line 10, a hybrid join search algorithm is processed. The hybrid join search algorithm consists of two parts: result segmentation and join operation. If there are two or more search results in the ResultList, the complex query statement will be split and combined into multiple result pairs. For instance, the complex query type Q can be split and combined into two result pairs, i.e., star((?s, $p_1$, $o_1$) ⋈ (?s, $p_2$, $o_2$)) and chain((?s, $p_3$, ?o) ⋈ ($s_4$, $p_4$, ?o)). In line 13, we initialize ResultList before the join operation to ensure that its query result is up-to-date. From lines 14 to 19, we perform the join operation for each result pair; if it is star type, then the hash join algorithm is used, otherwise the nested loop join algorithm is applied. This looping operation is done until the final result is obtained. After obtaining the final results, a recursive uniform resource identification (URI) lookup process is performed using link traversal techniques.

---

**Algorithm 1.** SPARQL join query algorithm for semantic search.

---

1:   # Example: a complex query type Q = $(?s, p_1, o_1) \bowtie (?s, p_2, o_2) \bowtie (?s, p_3, ?o) \bowtie (s_4, p_4, ?o)$
2:   **If** Q is a standard SPARQL triple pattern
3:      **For each** subquery in Q
4:       **If** distance between subquery and centroids in $k$-d tree is shortest **Then**
5:         Search the corresponding R*-tree by mapping list
6:         Adding search results to ResultList
7:       **Else** break
8:      **End For**
9:   **End If**
10: # Hybrid join search algorithm
11: **While** (two or more search results in ResultList)
12:     ResultList is split and combined into multiple result pairs
13:     Initialize ResultList
14:     **For** each result pair
15:      **If** result pair is star type **Then**
16:       ResultList ← Procedure HASH_JOIN(result pair)
17:      **Else**:
18:       ResultList ← Procedure NESTED_LOOP_JOIN(result pair)
19:     **End For**
20: **End While**

---

## 4. Experimental Evaluation

Based on experiments, we compared the proposed method with some existing popular methods. We should consider the issue of how to select the most suitable strategy for storing RDF data. For this, we investigated the balancing between two performance metrics, i.e., query response time and index load time [30]. The objective of the experiments was to show that we could achieve excellent join query performance with retrieval accuracy. We compared the learned semantic index structure, which we refer to as LearnS, with Quad [31], Darq [32], Dams [33], Midas [34], and Two-step [24]. Quad is an existing centralized approach, and Darq is a distributed approach; Dams is a recently proposed distributed approach, which combines adaptive hashing and the master-slave model; Midas is an index-based approach; and Two-step is a hybrid approach combining R*-tree and $k$-d trees. This evaluation is conducted on a server system with a 3.6-GHz Intel i7 CPU and 8 GB of memory. All programs were written in Python and Java languages on a server running Windows 10.

We used the LUBM (Lehigh University BenchMark) dataset [35] to obtain realistic results. The LUBM dataset is the most widely used benchmark dataset in the semantic web community. The LUBM dataset contains 230,061 triples, 38,334 subjects, 17 predicates, and 29,635 objects, and its size is 36.7 MB. The LUBM dataset offers well-designed test queries that fully cover features of a traditional reasoning system. However, LUBM does not support all join query types, so we extended the original version to support above seven different join types. We used 12 different benchmark join queries provided by [36], and we additionally made two chain and one cycle benchmark join queries based on the LUBM dataset. Thus, we provided 15 different join queries supporting all join query types. Among the 15 queries, 6 queries (Q1, Q3, Q4, Q5, Q11, and Q13) are star queries. Chain (Q6 and Q10), directed cycle (Q2), cycle (Q15), tree (Q9), single (Q14), and complex (Q7, Q8, and Q12) queries are 2, 1, 1, 1, 1, and 3, respectively. Additionally, we considered the LinkedGeoData dataset [37] to provide a more solid experiment. LinkedGeoData is a large spatial knowledge base, which has been derived from OpenStreetMap for Semantic Web. It interlinks geo-data with other knowledge bases in the LOD cloud. LinkedGeoData contains 2,207,295 triples, 552,541 subjects, 1320 predicates, and 1,308,214 objects. Its size is 327 MB.

### 4.1. Join Query Performance

To evaluate the join query performance of the proposed learned semantic (LearnS) index structure, we performed 15 SPARQL join queries that expressed the above seven join query types. Figure 4 illustrates the join query performance for various join query types. Experiment results show that the query performance of LearnS was always superior to those of other methods. This is mainly due to the learned indexing structure, which prunes a lot of unnecessary data scans during the join query. But we can see that Q4 approach performance was exceptionally slightly worse than Quad. The Darq and Midas approaches performed considerably worse than the Quad, Dams, Two-step, and LearnS approaches because of considerably larger intermediate results and more time for I/O and CPU. Although Darq and Dams are distributed approaches, Dams adopts the distributed semi-join query to minimize join query numbers without inter-communicating with endpoints, so it greatly reduces the communication cost. This characteristic has also been verified in the LinkedGeoData dataset. The Quad approach showed slightly worse performance than the Dams, Two-step, and LearnS approaches because Quad needs to load six B+-trees to support all access query patterns and decompress intermediate results in a short time. However, the Two-step approach considers only an R*-tree with *k*-d trees without storing all possible access combinations. The Two-step approach performed worse than LearnS because it is based on the hash function that does not describe exact coordinates in the multidimensional space; it contains a number of false hits not fulfilling the query condition. However, the LearnS approach turns heterogeneous hash codes into homogeneous sets of the adjacent embedding space. It further reduces the intermediate results using the clustering techniques described in Section 3.2.

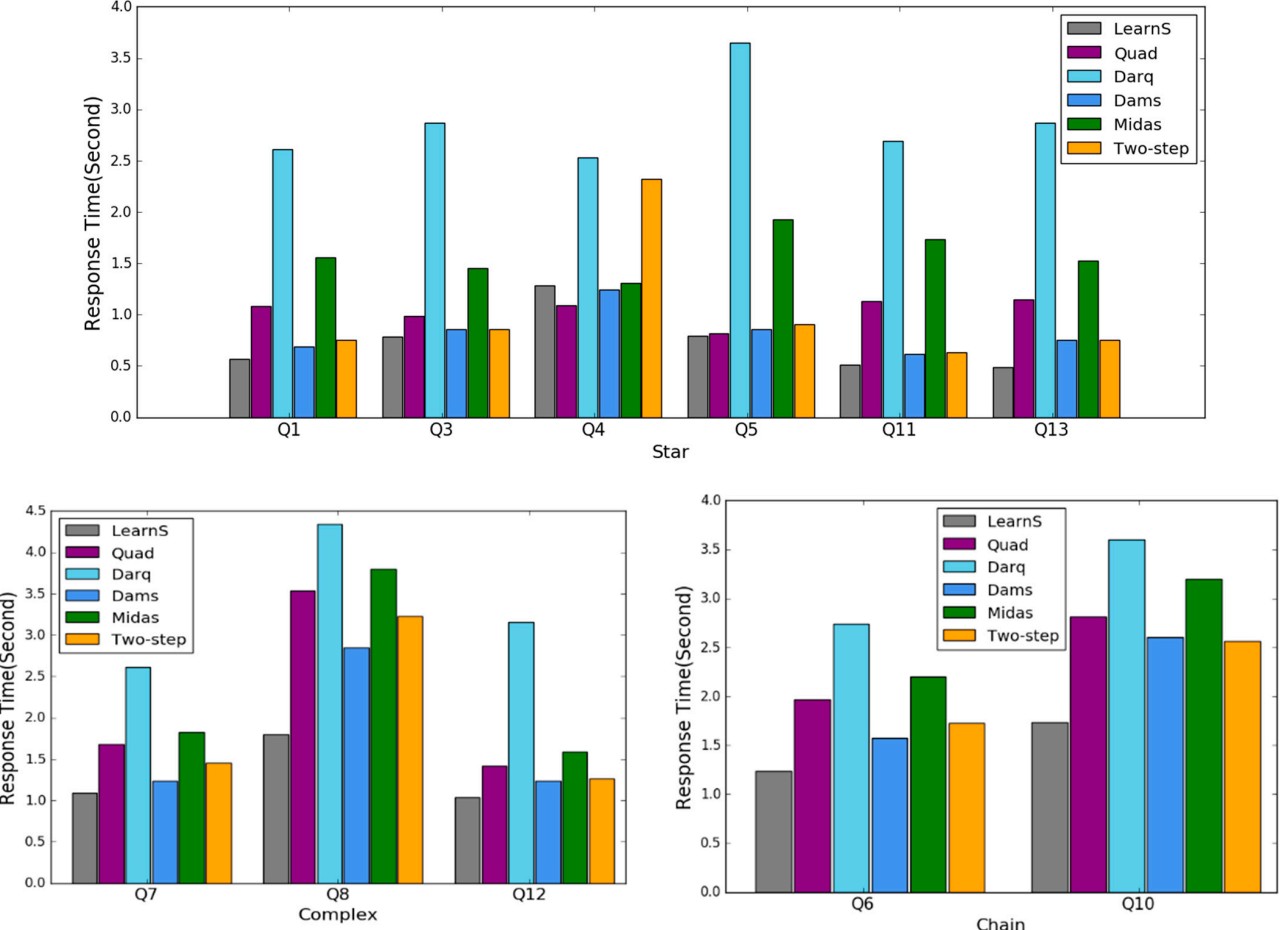

**Figure 4.** *Cont.*

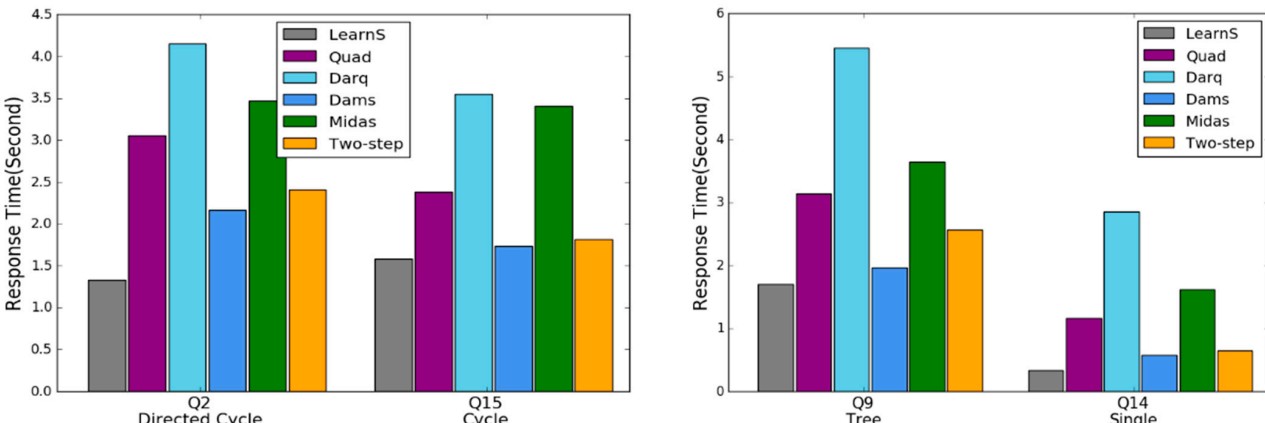

**Figure 4.** Join query performance for various join query types on LUBM dataset.

Figure 5 shows the join query performance on the LinkedGeoData dataset. In this figure, join query types only focus on the most common star, complex, chain, and cycle types. This figure shows that experimental results for LinkedGeoData are almost identical to the experimental results for LUBM. Under the LUBM dataset, the join query performance of the LearnS, Dams, and Two-step approaches achieved better results than those of the other methods. Similarly, under the LinkedGeoData dataset, the query performance of LearnS always beat the performance of other methods. However, the performance gap between LearnS and Two-step was particularly prominent in the chain type.

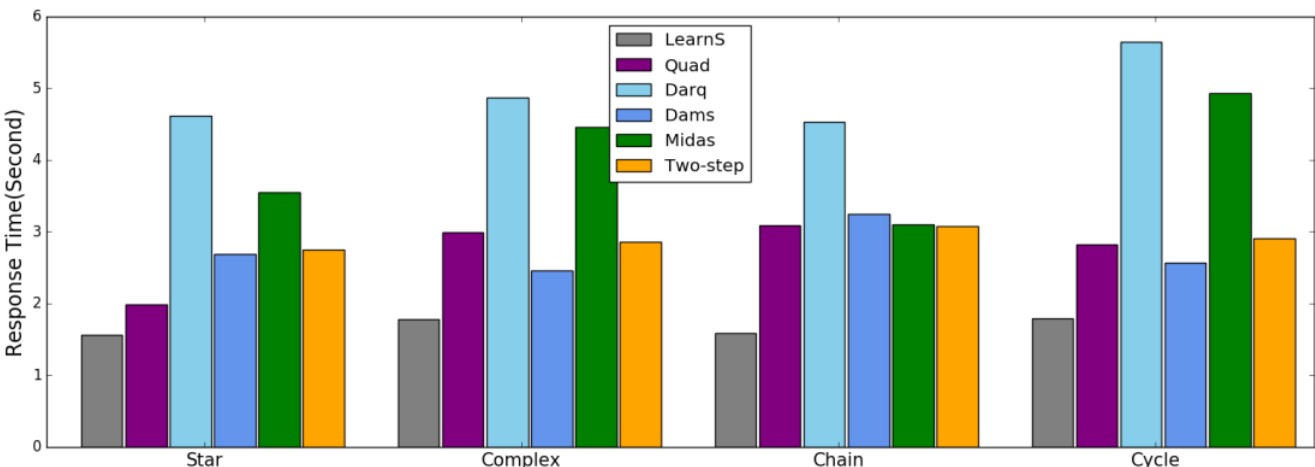

**Figure 5.** Join query performance for various join query types on LinkedGeoData dataset.

### 4.2. Index Building Time

The efficiency of the LearnS approach is not only reflected in join search but also in the building time. Figure 6 shows the building time of mainstream index structures. As Quad needs to build six B+ tree indexes, its time cost is the highest. The Two-step approach adopts the combination of R*-tree and *k*-d trees. Due to its hash-based semantic representation, the space complexity of indexing is higher than LearnS. Hence, it needs more building time than LearnS. The Midas and Darq approaches are index-based and distributed retrieval systems. However, the former uses *k*-d tree as its main index structure, hence its index building time is significantly lower than the latter. Dams, compared to Midas, spends more time because it does hash partitioning RDF data into multiple slave nodes.

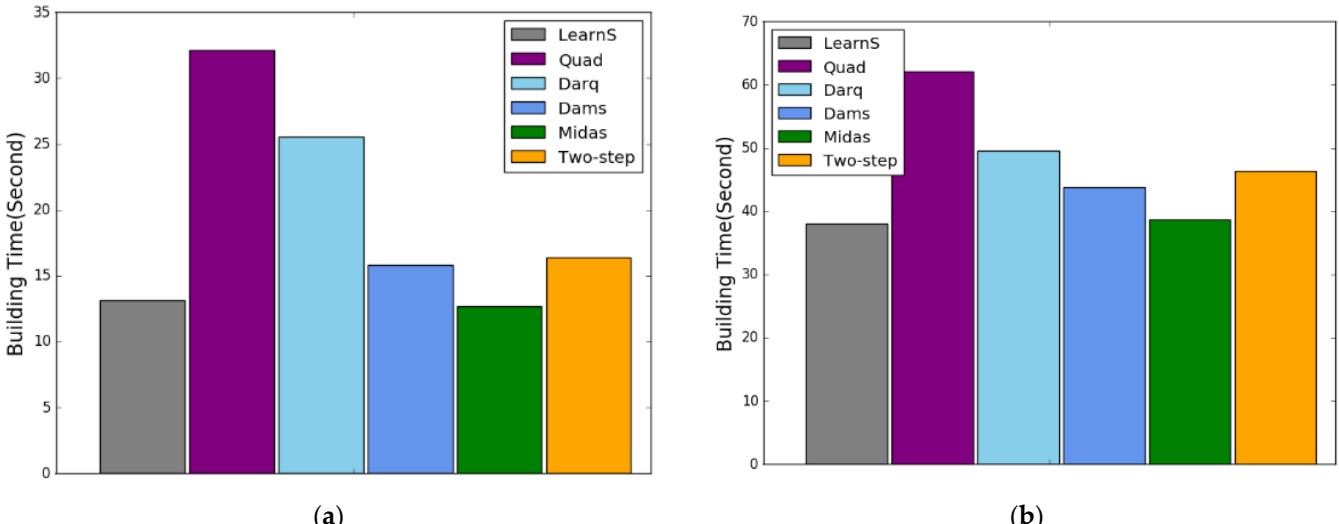

**Figure 6.** Building time for various index structures. (**a**) LUBM dataset, (**b**) LinkedGeoData dataset.

### 4.3. Semantic Clustering Performance

We measured the semantic clustering performance by employing Recall (*R*), Precision (*P*), and F-measure (*F*). The measure *R* is a measure of completeness, whereas *P* is a measure of exactness or fidelity. An inverse relationship exists between *R* and *P* wherein it is possible to increase only one at the cost of reducing the other. Usually, *R* and *P* measures are not discussed in isolation; both are combined into a single measure, such as *F*, which is the weighted harmonic mean of *R* and *P*. We consider these measures for semantic search. Let Ω be the set of relevant resources, *C* (correct) be the number of returned relevant resources, *I* (incorrect) be the number of returned irrelevant resources, and *M* (missing) be the number of missing relevant resources. We define:

$$R = \frac{C}{C+M}, P = \frac{C}{C+I} \text{ and } F = \frac{2P \times R}{P+R} \tag{5}$$

where the size of Ω is equal to *C* + *M*.

Table 2 and Figure 7 enumerate the semantic clustering performance for various embedding methods in the case of random semantic search. We conducted an additional experiment focusing on the performance of clustering algorithms, such as DBSCAN and *K*-means. The *K*-means clustering algorithm is mainly applicable to spherically distributed data. The experiment results showed that the translation-based models—TransE and TransR—were better than the bilinear-based models—RESCAL, DistMult, and complex—and the neural network-based model—ConvE. In particular, our LearnS demonstrates excellent performance on *P*, *R*, and *F*. The main reason is that this model solves the problem of annotation heterogeneity by projecting entities from entity space to corresponding relation space and applying the density-based spatial clustering algorithm. In general, there was an inverse relationship between recall and precision. In Figure 7, however, TransH, RESCAL, DistMult, and complex show poor results for both. This figure shows that ConvE also has a slightly poor performance in these measures. Although the ConvE model has shown its superiority in knowledge graph completion by using two-dimensional convolution techniques to extract the feature vector of triples, the semantic clustering effect is unsatisfactory. From Table 2, we can also observe that the DBSCAN method exhibit better performance than the *K*-means method. Figure 8 shows that all LUMB data are converted into the three-dimensional space. The TransR and LearnS models showed the excellent clustering performance compared with other methods.

**Table 2.** Evaluation of semantic clustering performance.

| Models | Precision (*P*) DBSCAN (*K*-Means) | Recall (*R*) DBSCAN (*K*-Means) | F-Measure (*F*) DBSCAN (*K*-Means) |
|---|---|---|---|
| TransE | 0.335 (0.250) | 0.903 (0.865) | 0.489 (0.388) |
| TransH | 0.130 (0.060) | 0.240 (0.080) | 0.168 (0.068) |
| TransR | 0.365 (0.340) | 0.955 (0.901) | 0.511 (0.402) |
| RESCAL | 0.280 (0.250) | 0.220 (0.182) | 0.246 (0.210) |
| DistMult | 0.210 (0.334) | 0.160 (0.287) | 0.182 (0.308) |
| complex | 0.325 (0.310) | 0.221 (0.268) | 0.263 (0.287) |
| ConvE | 0.340 (0.335) | 0.480 (0.425) | 0.398 (0.375) |
| LearnS | 0.386 (0.345) | 0.970 (0.930) | 0.552 (0.503) |

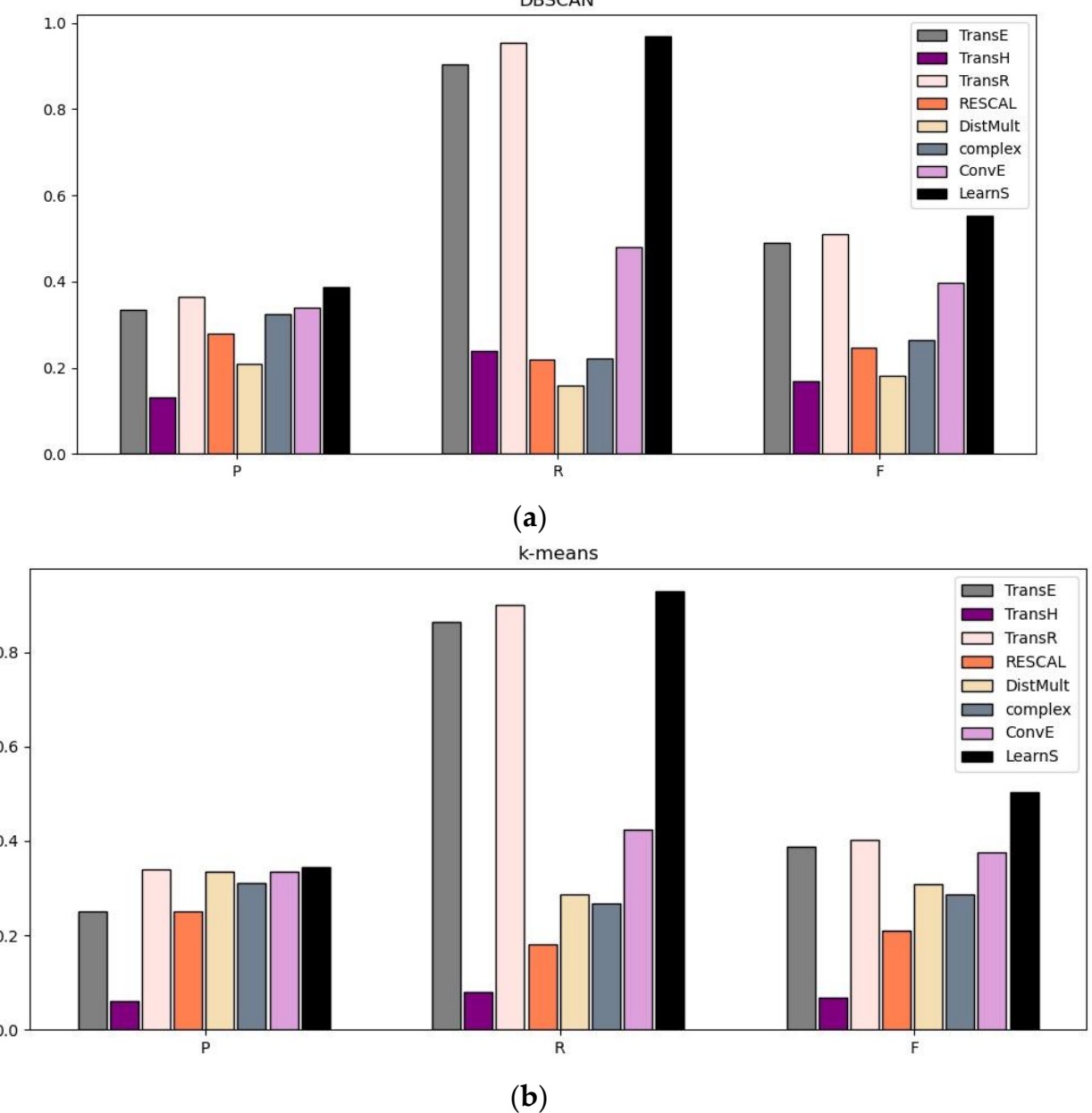

**Figure 7.** Semantic clustering performance based on DBSCAN and *K*-means. (**a**) Semantic clustering performance based on DBSCAN. (**b**) Semantic clustering performance based on *K*-means.

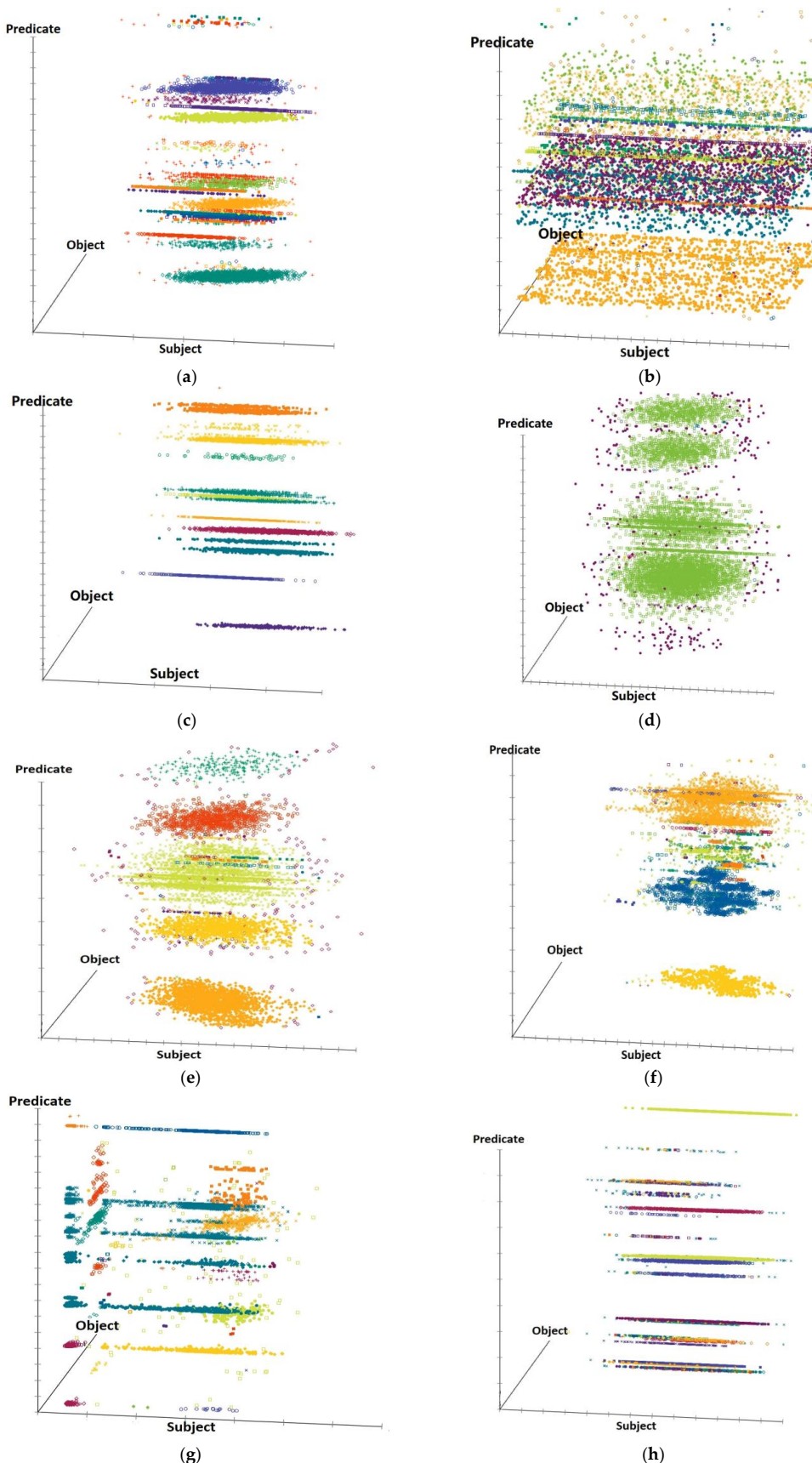

**Figure 8.** Visualization of the embedded LUMB dataset. (**a**) TransE, (**b**) TransH, (**c**) TransR, (**d**) RESCAL, (**e**) Complex, (**f**) DistMult, (**g**) ConvE, (**h**) LearnS.

## 5. Conclusions and Future Work

Efficient query processing for the LOD cloud is one of the most challenging requirements in the semantic web community. In this paper, we have proposed a learned semantic index structure to semantically store, index, and query LOD. We presented a SPARQL join query algorithm based on the learned index structure for semantic search. In addition, we proposed the extension of the DBSCAN and NN algorithm. Our index structure consists of two levels—one for determining the target cluster (i.e., $k$-d tree) and the other for searching final results (i.e., R*-trees). Contrary to several earlier investigations on this subject, which only considered applying hash functions, we consider embedding and clustering techniques. Searching approximate NNs is to find $k$ approximate nearest vectors in embedding space to a query pattern by using $k$-d tree. The experiment results demonstrate that our method significantly improves the join query performance compared with state-of-the-art LOD index structures. As future work, it is desirable to discuss in detail the construction cost of our index structure in terms of time, memory, and storage space. The maintenance cost due to update operations also should be discussed. We will be also focused on the investigation of precision and recall performance for the other various index structures.

**Author Contributions:** Writing—original draft preparation, Y.S.; writing—review and editing, S.-J.C.; writing—review and editing, Y.L.; funding acquisition, Y.L.; project administration, Y.L. All authors have read and agreed to the published version of the manuscript.

**Funding:** This research was supported by the Basic Science Research Program through the National Research Foundation of Korea (NRF) funded by the Ministry of Education (No. 2016R1D1A1B02008553). This study was supported by the BK21 FOUR project (AI-driven Convergence Software Education Research Program) funded by the Ministry of Education, School of Computer Science and Engineering, Kyungpook National University, Korea (4199990214394).

**Institutional Review Board Statement:** Not applicable.

**Informed Consent Statement:** Not applicable.

**Data Availability Statement:** Not applicable.

**Conflicts of Interest:** The authors declare no conflict of interest.

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
