# Peer review of "Learned Semantic Index Structure Using Knowledge Graph Embedding and Density-Based Spatial Clustering Techniques"

_applsci, doi:10.3390/app12136713_

Round 1

Reviewer 1 Report

The work proposes a learned index structure using knowledge graph embedding and spatial clustering techniques for semantic search.
The proposed index structure consists of two levels, one for determining the target cluster based on k-d tree, and the other for searching final results based on R*-trees.

The work is well presented. However, more recent works and a larger  experiment are required.

line 43: "For example, “beverage” and “fruit-juice” have the same semantics". 
I would say that this is true "in certain contexts". Indeed, in general, it is not true.

lines 78-79: "To the best of our knowledge, we are the first to present a learned index structure of the LOD cloud using KGE models". It is not clear what this sentence wants to convey. In fact, if by "LOD cloud" you mean the 1,301 datasets, this is not reflected in the experimentation which involves the LUBM dataset and then a small number of triples. 

lines 289-290: Please, can you explain why different join algorithms are used? 

line 295: In the example of query Q, it should be better indicate different names for different variables and uris. As it is, the first and the second triples in the query Q are identical.  

line 311: concerning the indexing methods selected from the literature, three out of four have been proposed before the 2011. Please, consider the following for more recent works:
Tanvi Chawla, Girdhari Singh, Emmanuel S. Pilli, Mahesh Chandra Govil:
Storage, partitioning, indexing and retrieval in Big RDF frameworks: A survey. Comput. Sci. Rev. 38: 100309 (2020) 

Algorithm 1: it is not clear if the algorithm is for any kind of query pattern or just for the pattern Q. For instance, what happens at line 15 and 16 if the query is the conjunction of five triples instead of four? 

line 324: I would encourage to provide a more solid experiment by considering:
- more and larger rdf datasets (e.g., Yago);
- a larger number of queries for each type.

lines 342-344: "Two-step performs worse than LearnS because it is based on the hash function that does not describe exact coordinates in the multidimensional space; it contains a number of false hits not fulfilling the query condition". 
According to that, the performances of Two-steps are discussed not only about "response time" (efficiency), but also about efficacy (e.g., precision). What about precision and recall of the other methods? 

TYPOS
lines 176-177: desity-based -> density-based

lines 423-430: Please, fix the "Autor Contributions" paragraph

Reviewer 2 Report

This paper presented an interesting approach on semantic indexing with KG Embedding by applying and upgrading the ideas shown in [24], [10] and other papers.    The work has been evaluated on three aspects, query performance, index building time, and semantic clustering performance. The first two were related to their overall query processing performance in time and the final one was for "embedding performance" which can be seen as how they can gather "similar" data by them. According to the given results, the proposed approach produced better performance on these compared existing approaches.   However, the paper did not clearly mention that they (the compared approaches) were still valid for these evaluations as representatives of "state-of-the-art" approaches. In this area we are having very intensive efforts in its development and even for recent years we have some important surveys in the area which were not referred in this paper(e.g., [A],[B],[C],[D]). Authors should clarify this point by referring these surveys and should justify how their choice on the evaluations were appropriate. 

  [A]Marcin Wylot, Manfred Hauswirth, Philippe Cudré-Mauroux, and Sherif Sakr. 2018. RDF Data Storage and Query Processing Schemes: A Survey. ACM Comput. Surv. 51, 4, Article 84 (July 2019), 36 pages. https://doi.org/10.1145/3177850  

[B]Ali, W., Saleem, M., Yao, B. et al. A survey of RDF stores & SPARQL engines for querying knowledge graphs. The VLDB Journal 31, 1–26 (2022). https://doi.org/10.1007/s00778-021-00711-3  

[C]Ben Mahria, B., Chaker, I. & Zahi, A. An empirical study on the evaluation of the RDF storage systems. J Big Data 8, 100 (2021). https://doi.org/10.1186/s40537-021-00486-y

  [D]S. Ji, S. Pan, E. Cambria, P. Marttinen and P. S. Yu, "A Survey on Knowledge Graphs: Representation, Acquisition, and Applications," in IEEE Transactions on Neural Networks and Learning Systems, vol. 33, no. 2, pp. 494-514, Feb. 2022, doi: 10.1109/TNNLS.2021.3070843.

Round 2

Reviewer 1 Report

Here some 2 comments:

Lines 43-44: "For example, “vehicle” and “means_of_transport” have the same semantics but cannot be matched because they do not belong to the same synonym set".
I still do not agree with this sentence. In fact, if two terms have the same semantics then they are synonyms.

Lines 304-305: Still not clear the expression (?s, p, o)⋈(?s, p, o). As they are written, the two triples are the same, so, what the join needs for? In the star pattern, the subject of the two triples must be the same (and this is why the variable ?s is the same), but the predicates and the objects should not both coincide. In order to avoid misunderstandings, why don't you use (?s, p1, o1)⋈(?s, p2, o2)?

Author Response

We thank anonymous review for their helpful suggestions and comments.

Lines 43-44: "For example, “vehicle” and “means_of_transport” have the same semantics but cannot be matched because they do not belong to the same synonym set".
I still do not agree with this sentence. In fact, if two terms have the same semantics then they are synonyms.

This is modified in page 1 (in row 43) (revised as below).

For example, “Benz E-250” and “Mercedes Benz E class 250” have the same semantics but cannot be matched because they do not belong to the same synonym set.

Lines 304-305: Still not clear the expression (?s, p, o)⋈(?s, p, o). As they are written, the two triples are the same, so, what the join needs for? In the star pattern, the subject of the two triples must be the same (and this is why the variable ?s is the same), but the predicates and the objects should not both coincide. In order to avoid misunderstandings, why don't you use (?s, p1, o1)⋈(?s, p2, o2)?

This is explained in page 7 (in row 297) and in page 8 (in row 305-306).

Reviewer 2 Report

The authors updated according to the comments.

It would be nice when the authors also refer some other recently-published surveys to make it clear that the authors' approach has been compared to the state-of-the-art in the field.

Author Response

We thank anonymous review for their helpful suggestions and comments.

It would be nice when the authors also refer some other recently-published surveys to make it clear that the authors' approach has been compared to the state-of-the-art in the field.

This is explained in page 7 (in row 284-285) (revised as below).

There are two key points for complex JOIN queries: join ordering and selecting the most suitable join algorithm [29].

We refer other recently-published survey [29] to make clear that our approach has been compared to the state-of-the-art in the field.